# Is Micro X-ray Computer Tomography a Suitable Non-Destructive Method for the Characterisation of Dental Materials?

**DOI:** 10.3390/polym13081271

**Published:** 2021-04-14

**Authors:** Andreas Koenig, Leonie Schmohl, Johannes Scheffler, Florian Fuchs, Michaela Schulz-Siegmund, Hans-Martin Doerfler, Steffen Jankuhn, Sebastian Hahnel

**Affiliations:** 1Department of Dental Prosthetics and Materials Science, Leipzig University, 04103 Leipzig, Germany; Leonie.Schmohl@medizin.uni-leipzig.de (L.S.); Florian.Fuchs@medizin.uni-leipzig.de (F.F.); hahnel@medizin.uni-leipzig.de (S.H.); 2Institute of Chemical Technology, Leipzig University, 04103 Leipzig, Germany; j.scheffler@studserv.uni-leipzig.de; 3Institute of Pharmacy, Pharmaceutical Technology, Leipzig University, 04317 Leipzig, Germany; schulz@uni-leipzig.de; 4Department of Mechanical and Energy Engineering, University of Applied Sciences, 04277 Leipzig, Germany; hans-martin.doerfler@htwk-leipzig.de; 5Felix Bloch Institute of Solid State Physics, Leipzig University, 04103 Leipzig, Germany; jankuhn@uni-leipzig.de

**Keywords:** dental materials, ionising radiation, micro-CT, µXCT, X-ray radiation, DSC, RBC, PMMA, Harvard Cement

## Abstract

The aim of the study was to investigate the effect of X-rays used in micro X-ray computer tomography (µXCT) on the mechanical performance and microstructure of a variety of dental materials. Standardised bending beams (2 × 2 × 25 mm^3^) were forwarded to irradiation with an industrial tomograph. Using three-dimensional datasets, the porosity of the materials was quantified and flexural strength was investigated prior to and after irradiation. The thermal properties of irradiated and unirradiated materials were analysed and compared by means of differential scanning calorimetry (DSC). Single µXCT measurements led to a significant decrease in flexural strength of polycarbonate with acrylnitril-butadien-styrol (PC-ABS). No significant influence in flexural strength was identified for resin-based composites (RBCs), poly(methyl methacrylate) (PMMA), and zinc phosphate cement (HAR) after a single irradiation by measurement. However, DSC results suggest that changes in the microstructure of PMMA are possible with increasing radiation doses (multiple measurements, longer measurements, higher output power from the X-ray tube). In summary, it must be assumed that X-ray radiation during µXCT measurement at high doses can lead to changes in the structure and properties of certain polymers.

## 1. Introduction

Micro X-ray computer tomography (µXCT) is regarded as a powerful, non-destructive imaging technique that can be used for the visualisation and analysis of the 3D structure of non-living objects (in vitro) or small animals. As a result of the high radiation exposure and the distinct measurement setup (object rotates and must not move), the technique cannot be applied to humans, e.g., in in vivo studies. In the last twenty years, µXCT has gained wide acceptance in medical, dental, and materials technology, as well as science (Figure 1).

Swain and Xue [1] categorise the different clinical topics in which the method can be used in dentistry:▪Enamel thickness measurement and tooth characterisation.▪Analysis of root canals.▪Craniofacial skeletal development and structure.▪Analysis of biomechanical behaviour (in combination with finite element modelling (FEM)).▪Tissue engineering.▪Quantification of mineral concentration in teeth.▪Implant and peri-implant bone analysis.

In addition, µXCT may be applied in dental materials science, including in the analysis of air voids [2,3], filler distribution [4], investigations regarding polymerisation shrinkage [5,6,7,8,9,10], or the identification of microcracks within dental materials [11].

Electromagnetic radiation includes different ranges regarding its frequency and wavelength, including, for instance, ultraviolet, X-ray, or gamma radiation.

The ionising radiation range in which electrons are released from atoms or molecules commences in the short-wave ultraviolet range at <250 nm. Ionising radiation can either lead to physically induced cross-linking or to degradation of polymers [12].

UV radiation in the non-ionising range (usually > 340 nm) is currently used in the dental sector in vitro, e.g., for post-polymerisation of printable monomers (Lin et al. 2020; Assaf et al. 2020; Gonçalves et al. 2010). In the mid-1960s and 1970s, it was also used in vivo [13] for the polymerisation of polymer-based materials. Thereafter, the curing composite initiation was changed from UV light curing to visible light curing.

Due to the use of photoinitiators in the monomers [14], the harmful effect of the ionising radiation (electromagnetic, but also particle radiation), and its minimal penetration depth when using radiopaque fillers in the monomer [8,15], ionising radiation has no relevance to clinical dentistry, in contrast to the field of science (Figure 1) [16,17,18,19]. 

In other technical areas, the cross-linking of polymers such as polyethylene (PE) or polyamides (PAMs) is increased by the free radicals produced by ionising radiation. As a result of this treatment, the mechanical properties of the materials, as well as their chemical and thermal resistance, improve [20]. However, similar irradiation treatments may also deteriorate the properties of other polymers, such as an increase in brittleness (e.g., of polypropylene (PP)) (possible application: splinter-free ampoule skewers [20]) or a change in melting behaviour (possible application: increasing the fire resistance of high-performance concrete [21]). These considerations underline that ionising radiation may change the internal structure of a material and affect its properties, as observed in polymer materials.

Against this background, the aim of the current study was to investigate the effect of X-rays associated with µXCT investigations on the performance and structure of a selection of dental materials. The null hypothesis was that X-ray radiation associated with µXCT has no influence on the mechanical performance and internal structure of the materials.

## 2. Materials, Preparation and Methods

### 2.1. Materials

Five dental materials and one non-dental material were selected for the current investigation. The product information retrieved from the technical datasheets, as well as the products’ batch numbers, is displayed in Table 1.

### 2.2. Preparation

Bending beams (2 × 2 × 25 mm^3^) were produced using material-specific processes. Only beams without apparent defects such as cracks or blowholes on their surface were forwarded to irradiation and further analysis. At least twelve beams of each material were irradiated, and as a reference group twelve beams of each material were not irradiated. Both groups were tested for flexural strength and subsequently analysed with differential scanning calorimetry (DSC).

Harvard Cement (HAR) was mixed by hand from 2.5 g oxide powder and 45 drops of phosphoric acid solution on a cooled glass plate. Bending beams were produced in accordance with DIN EN ISO 4049 by condensing the uncured cement into a Teflon formwork. The bottom and top sides of the formwork were covered with polyethylene (PE) films and tightly screwed. After five minutes, the formwork was carefully stripped.

Standardised blocks (25.0 × 25.0 × 12.5 mm^3^) were automatically milled from blanks (Ø/h = 98.4/20 or 26 mm) of the indirect resin-based composite Structur CAD (STC) and the poly(methyl methacrylate) (PMMA)-based material VITA Vionic Base (VIO) using a five-axis dental milling machine (inLab MC X5; Dentsply Sirona Deutschland GmbH, Bensheim, Germany). The PMMA-based material IvoBase Hybrid (IVO) was pressed into a preproduced blank and polymerised. Subsequently, standardised blocks (25 × 25 × 12.5 mm^3^) were milled from this blank employing the above milling procedure. Finally, bending beams were cut from the standardised blocks in a two-step process using a precision saw (IsoMet^®^ 4000 Linear Precision Saw, Buehler, IL, USA).

Bending beams (2 × 2 × 25 mm^3^) were produced from IMPRIMO LC model (IMP) with a digital light processing (DLP) 3D printer (Asiga Max, Scheu-Dental GmbH, Iserlohn, Germany). Bending beams (2 × 2 × 25 mm^3^) were produced from the non-dental material polycarbonate with acrylnitril-butadien-styrol (PC-ABS) model (PCM) using a fused deposition modelling (FDM) 3D printer (FDM Titan, Stratasys Ltd., Eden Prairie, MN, USA).

### 2.3. Methods

#### 2.3.1. Differential Scanning Calorimetry (DSC)

Differential scanning calorimetry (DSC) was used to identify changes in the internal structure of the materials that were potentially caused by X-rays during µXCT. Following the bending test, a sub-sample of each material was cut from the centre of the beams (approx. 9 mg), placed in an aluminium pot, and sealed with an appropriate lid (THEPRO GbR, Heinsberg, Germany). In the case of HAR, manually perforated lids were used to prevent bursting due to vapour pressure. DSC was performed with a Polymer DSC R and a TSO801RO sample robot with STARe software 14.0 (Mettler Toledo GmbH, Gießen, Germany). An empty, sealed pot was used for reference. With the exception of HAR (600 °C), the maximum possible temperature (T_max_) below the onset of decomposition was selected (IMP 345 °C, IVO 260 °C, PCM 380 °C, STC 300 °C, VIO 270 °C). Using nitrogen with a flow rate of 40 mL/min as inert purge gas, the following temperature programme was applied twice in a row: 5 min at 25 °C isothermal segment, heating to T_max_ at a rate of 10 K/min, followed by 5 min isothermal segment, and cooling to 25 °C at a rate of 10 K/min. The characteristic glass transition temperatures were determined using the midpoint ISO method of the STARe software.

#### 2.3.2. Micro X-ray Computed Tomography (µXCT)

A microfocus X-ray computed tomograph (industrial tomograph) was used to irradiate the specimens and to identify local discontinuities. An FXE 225.99 X-ray tube (focal spot diameter 0.6 µm, tungsten target) fabricated by YXLON International GmbH (Hamburg, Germany) and a 1621xN 2D detector (2048 × 2048 pitches, CsI, pitch size 200^2^ µm^2^) fabricated by PerkinElmer (Waltham, MA, USA) were used. Radiation dose was estimated on the basis of Zhao et al. [22] for the measurement setup with approx. 52 Gy. The settings employed in the current study are displayed in Table 2.

The three-dimensional datasets of 3/12 beams from each material were digitally cut and orientated with ImageJ (version 1.47, National Institutes of Health, Bethesda, MD, USA). For quantification of discontinuities, a “region of interest” (ROI) with the dimension (l/w/h = V: 1.87/1.87/11.45 = 40 mm^3^) was defined and used for the analyses with VGStudioMax (version 2.0, Volume Graphics GmbH, Heidelberg, Germany). The threshold was determined for the largest discontinuities based on the grey value distribution on the transition zone between material and discontinuities with ImageJ. A detailed description of the procedure has been reported by Koenig [23].

#### 2.3.3. Mechanical Tests

The three-point bending test for measuring the flexural strength of the specimens before (reference) and after irradiation was performed with beams (2 × 2 × 25 mm^3^) in accordance with DIN EN 841-1 and DIN EN ISO 4049 using a servomechanical testing machine (ZwickRoell GmbH & Co. KG, Retroline 10 kN, Ulm, Germany). The measurement setup had to be adapted according to the ductility and strength of the individual material type. The test settings are displayed in Table 3. 

The test was stopped when the load had dropped to 50% of the maximum load. The flexural strength (*σ**_F_*) was calculated using the the beam height (*h*), the beam width (*b*), the distance between the supports (*l*), and the maximal load (*F*) (Equation (1)):(1)$$\sigma_{F}=\frac{3Fl}{2bh^{2}}$$

In addition to flexural strength, the deformation at the point of maximum load was also determined. 

The single values of flexural strength in each group were analysed for normal distribution using the Shapiro–Wilk test. The Mann–Whitney test was employed for statistical analysis of not normally distributed datasets (HAR, IMP, IVO, STC) and the t-test was used for the analysis of normally distributed datasets (PCM, VIO). The level of significance was set to 0.05.

## 3. Results

### 3.1. Flexural Strength

The mineral-based material HAR showed linear elastic deformation bending stress behaviour, while the resin-based and polymer-based materials showed linear elastic and plastic deformation bending stress behaviours. IMP and VIO had a plastic post-fracture behaviour, PCM less post-fracture behaviour, and IVO and STC brittle fracture behaviour (Figure 2).

Means and standard deviations were calculated for flexural strength (Equation (1)). Flexural strength and deformation at the maximum load are displayed in Figure 3 and Figure 4. A statistically significant difference in flexural strength and deformation at the maximum load (*p* < 0.05) between specimens that had been exposed to X-ray irradiation during µXCT measurements and unexposed specimens was only identified for PCM (Figure 3 and Figure 4). The difference between both groups of PCM was highly significant (*p* < 0.01).

### 3.2. Microstructure

Differences in the microstructure of the various materials were visualised in cross sections and porosity values in the “region of interest” (ROI). In HAR, large single air voids and cracks connecting the air voids were identified. The milled “computer-aided design/computer-aided manufacturing” (CAD/CAM) materials IVO and STC featured the lowest porosity and the 3D-printed material PCM the highest porosity. The 3D-printed materials PCM and IMP showed a repeating pore structure, which was repetitively identified in the direction of the 3D-printing process (Figure 5).

### 3.3. Changes in the Internal Structure of the Materials

DSC curves of HAR showed no reversible effects. Only irreversible, large, broad, and overlapping endothermic signals could be detected (Table A1 (Appendix A). The peak temperatures of the first double signal (121 and 177 °C) and the second peak around 303 °C were consistent with the literature and can be attributed to decomposition resulting from the release of free and bound water [24]. Slight differences between irradiated and unirradiated samples can be attributed to differences in weight loss of the manually perforated crucible lids and complex rehydration processes within the different mineral phases [24].

The resin-based materials (IMP, SPC) exhibited no reversible glass transitions in the temperature regions analysed in the current study. In highly cross-linked polymers, the mobility of polymer chains is limited and no glass transitions occur. 

In PCM, the DSC curves of the first heating process show no notable differences between X-ray irradiated and unirradiated samples. The first endothermic peak and the two glass transitions around 105 °C and 130 °C fit the data issued by the manufacturers. PCM is a polymer blend consisting of polycarbonate and acrylonitrile-butadiene-styrene. As the components are only partially mixable, multiple glass transitions can be detected. The lower glass transitions (T_g_) can be assigned to glass transition of the styrene acetonitrile component of ABS and the higher T_g_ to polycarbonate (polybutadiene component T_g_ < −85 °C below measured temperature region) [25,26].

The T_g_ of PMMA-based materials (IVO, VIO), with and without X-ray exposure, showed no notable differences in the first heating curves. The T_g_ are in good agreement with values reported in literature [27]. In the second heating curves, the T_g_ of irradiated samples was slightly increased and about 10 K above the T_g_ of the unirradiated samples, with the latter clearly reduced (Figure 6).

## 4. Discussion

### 4.1. Harvard Cement (HAR)

With 27%, the highest percental standard deviation in flexural strength was identified for HAR. The irregularly distributed pores and cracks in the bending beams caused highly scattered results, especially in three-point bending tensile tests where the single maximum bending moment occurs only in the centre of the specimen. Mineral-based materials such as HAR or ceramics have a higher attenuation coefficient in comparison to polymer-based materials due to the high atomic mass of the incorporated elements and the high resistance to X-rays resulting from the high atomic bonding forces [8,15]. Minerals, especially phosphates, zirconia, and silicates, are considered chemically stable as well as radiation resistant and are regarded as potential materials to immobilise radioactive waste [28,29,30]. This circumstance might serve as an explanation for why no differences were identified in the current study regarding the mechanical behaviour or the network of HAR specimens exposed to X-ray and unexposed specimens.

### 4.2. PC-ABS Model Material (PCM)

In contrast, PCM featured the highest porosity (6.71 ± 0.17 vol%) of all materials investigated, which correlated with the lowest bulk density. These results underline the studies conducted by Popescu et al. [31], who quantified the air void content between 6.14 and 7.82 vol% for a similar product based on the same polymer mix. Dana et al. [32] showed that the printing trajectory and the printing speed had a relevant influence on anisotropic pore structure. In the current study, the pore structure was regular (Figure 5); thus, the percental standard deviation was not higher than in the other materials with lower porosity (Figure 3 and Figure 4). The significant decrease in flexural strength and deformation might be the result of changes in the polymer network [33]. It is well known that polycarbonate (PC) is sensitive to UV light (3.26–120 eV), which has a lower energy radiation than X-rays (>120 eV) [34].

As discussed in detail below, for degradation of the polymer network, e.g., main chain cleavages due to X-rays (similar to poly(methyl methacrylate), see next section), one would expect the glass transition temperature in DSC to be lowered compared to intact networks. In contrast, the first heating T_g_ of the styrene acetonitrile component of ABS and the T_g_ of polycarbonate showed no relevant changes that could result from irradiation. The T_g_ of the polybutadiene component lies outside the examined temperature range [25,26]. Thus, the observed differences in performance can be attributed to changes in this component. These results are consistent with the works of Wady et al., who showed that the polybutadiene fraction in ABS formed by fused filament fabrication is most susceptible to degradation by radiation [35].

The tensile stress in the cross section of the beam only has a height of 1 mm (half of the beam height) and reaches its largest value at the edge of the beam. If the tensile stress curve in the beam cross section is linear and the flexural strength decreases by 30 % (54.8 to 38.3 MPa), a height of 0.3 mm of the cross section in the beam no longer has a mechanical effect. This simple assumption illustrates the depth to which the radiation has degraded the material.

### 4.3. Poly(methyl methacrylate) (IVO, VIO)

The flexural strength of the poly(methyl methacrylate) (PMMA)-based materials IVO and VIO was not influenced by exposure to X-rays. However, radiochemical degradation of PMMA by X-rays is well known, e.g., from the field of X-ray lithography (a structuring method in semiconductor and microsystems technology) [33]. Using Fourier transform infrared spectroscopy (FTIR) and nuclear magnetic resonance, Choi et al. [36] and Moore and Choi [37] demonstrated that high-energy radiations such as X-rays can cause main chain cleavages and remove many ester groups much faster than low-energy radiation (e.g., UV). Using gel permeation chromatography (GPC) in comparison with a Monte Carlo simulation model, Yates and Shinozaki [38] showed that the cleavage process of the main chains occurs both during exposure to soft X-ray energy, and randomly. In summary, it is likely that the radiation power of a single µXCT examination in the current study was not sufficient to significantly influence the performance of PMMA. Fundamental research shows that the polymer network is destroyed with increasing dosage (e.g., following multiple measurements or when using high X-ray power) [33,36,37].

For degradation of the polymer network, e.g., main chain cleavages due to X-rays, one would expect the glass transition temperature in DSC (where amorphous thermoplastics get viscous) to be lower than that of an intact network as a result of greater mobility of the polymer chains. This relation between lower molecular weight and lower glass transition temperatures is well-known by the Flory–Fox equation [39].

In contrast to this, when comparing the respective PMMA-based materials with and without X-ray exposure, the glass transitions showed no relevant differences in first heating curves. From this point of view, it seems reasonable to assume that the X-rays have no influence on the respective polymer networks. However, this conclusion has to be rejected because the second heating curves indicate that T_g_ of irradiated samples was slightly increased and was about 10 K higher than T_g_ of unirradiated samples (Figure 6). 

Thus, the X-rays seem to have a stabilising effect on the polymer network when the latter are subjected to further heating. A common method of enhancing the properties of thermoplastics—especially temperature stability—is the modification of polymers through the application of ionising radiation [12,40]. Depending on the polymer used and the reaction conditions applied, multiple processes can be induced by radical formation in the polymer chains. These processes include degradation through chain breakage, cross-linking, and combinations [12]. On this basis, it seems reasonable to conclude that:▪a combination of chain breakage and cross-linking was induced by X-ray irradiation, but did not have a relevant impact on T_g_ in DSC curves of first heating (nor mechanics).▪during the first heating, loose ends of broken chains previously caused by X-ray irradiation reconnected. 

The resulting reduced chain mobility therefore caused minor increases (IVO 2 °C, VIO 6 °C) in T_g_ values in DSC second heating curves (cf. shift from grey dotted line in Figure 6). It is a well-known phenomenon that no additional reaction peaks have to be visible in DSC curves [27]. 

The decrease in T_g_ in unirradiated samples could be explained by the beginning of thermal degradation or lowered orientation in the sample [41,42].

### 4.4. Resin-Based Composite (STC) 

While the type of the polymer employed in Structur CAD (STC) is unknown, µXCT identified large filler particles with high radiopacity, which is typical for all dental resin-based composites (RBCs). Moreover, a very low porosity was identified, which is typical of CAD/CAM-processed materials [4,8,15,43]. The high radiopacity is equivalent to a rapid decrease in radiation exposure within the sample, which serves as an explanation as to why X-rays have little or no influence on the mechanical performance of these materials as a result of the attenuation (Figure 3).

For the 3D-printed material processed by DLP (IMP), very few defects within the material were identified in contrast to the material processed by FDM (PCM) (Figure 5), which correlates with lower bulk density and a higher X-ray attenuation coefficient in IMP in comparison to PCM. Similar to STC, no detailed information regarding the polymer type used in the materials (IMP) is issued by the manufacturer. However, constant bending strengths after exposure to X-rays indicate that resin-based materials processed by DLP are not relevantly affected by a single irradiation in µXCT examinations.

### 4.5. Resume

Micro X-ray tomography can be a suitable method for the non-destructive analysis of dental materials, yet it must be regarded that high doses (high X-ray tube output, long irradiation time) can lead to changes in the structure (see DSC results of PMMA) and properties (see flexural strength of PC-ABS) of selected polymers, especially for very small sample geometries. It can be concluded that:(1)The influence of X-ray radiation during µXCT measurement has to be verified beforehand for each polymer type.(2)The X-ray tube output and the measurement time have to be limited.

## 5. Summary 

Within the limitations of the study, the null hypothesis cannot be confirmed. However, the following conclusions can be drawn:(1)A significant change in flexural strength due to X-ray irradiation during a single µXCT measurement occurred only with polycarbonate with acrylnitril-butadien-styrol (PC-ABS). No changes were detected in zinc phosphate cement (Harvard Cement), poly(methyl methacrylate) (PMMA), methacrylate, or other resin-based composites (RBCs).(2)A shift in glass transition (T_g_) in both PMMA samples after repeated heating indicates a slight change in temperature resistance. The temperature resistance could be the result of degradation of the polymer network, e.g., main chain cleavages, which are re-cross-linked by repeated heating.(3)Further studies are necessary to investigate the influence of X-rays during µXCT measurements on the composition and performance of materials.

## Figures and Tables

**Figure 1 polymers-13-01271-f001:**
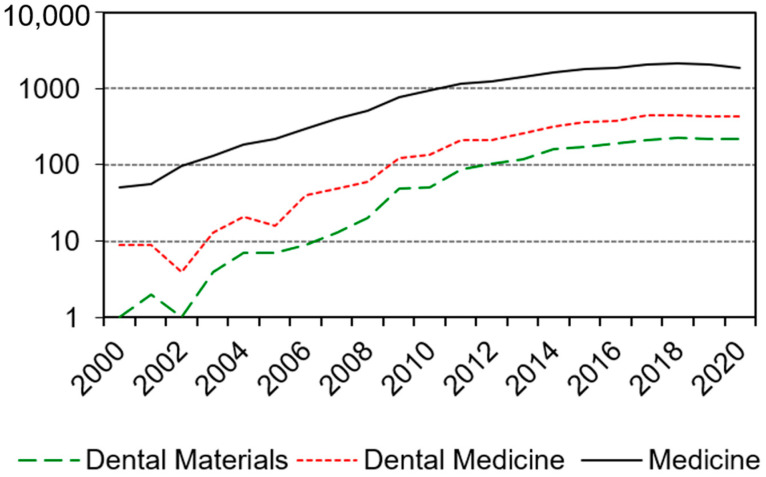
Hits identified on PubMed for the terms “microCT” or “micro X-ray tomography”.

**Figure 2 polymers-13-01271-f002:**
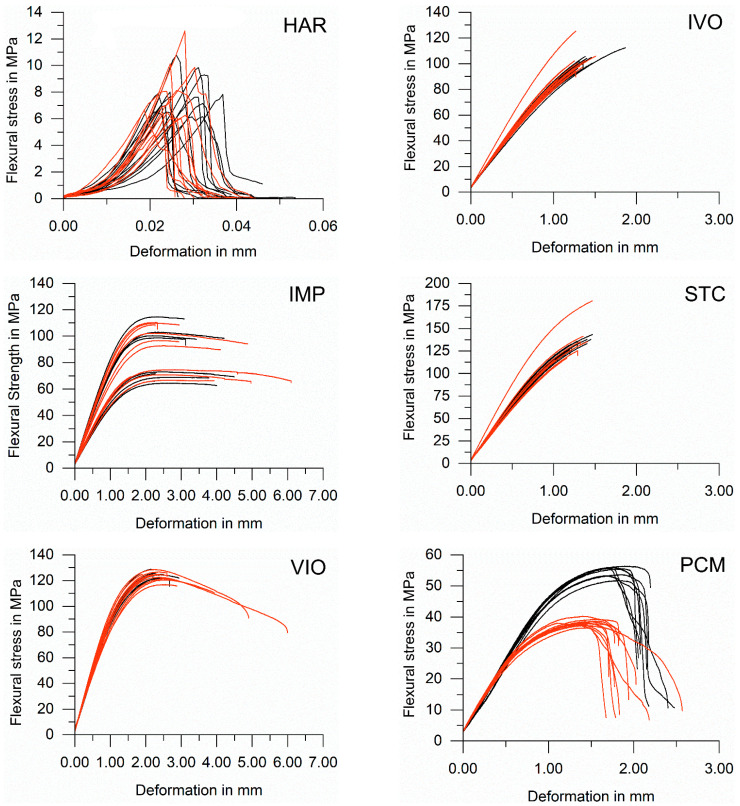
Deformation and flexural stress relations for materials exposed to X-ray irradiation during µXCT measurements (red) and unexposed specimens (black). Each curve describes the behaviour of a single sample. A total of 10 irradiated and 10 non-irradiated samples per material were tested.

**Figure 3 polymers-13-01271-f003:**
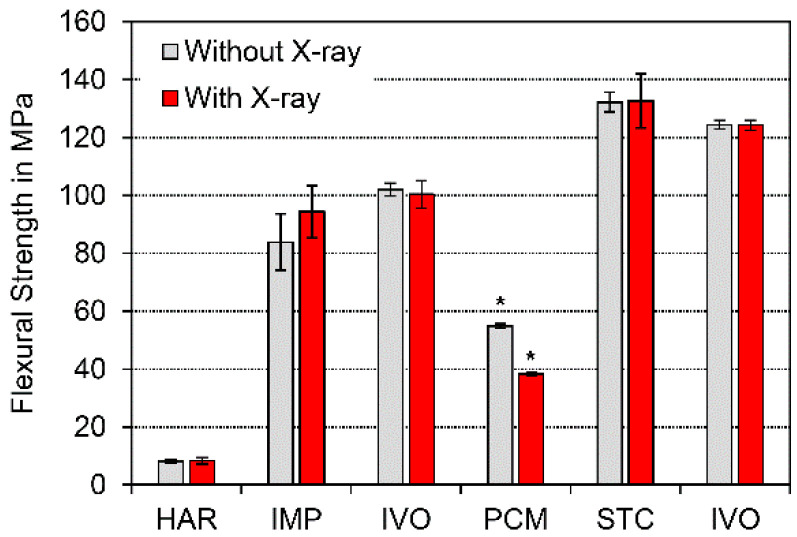
Flexural strength (*p* < 0.05 → *).

**Figure 4 polymers-13-01271-f004:**
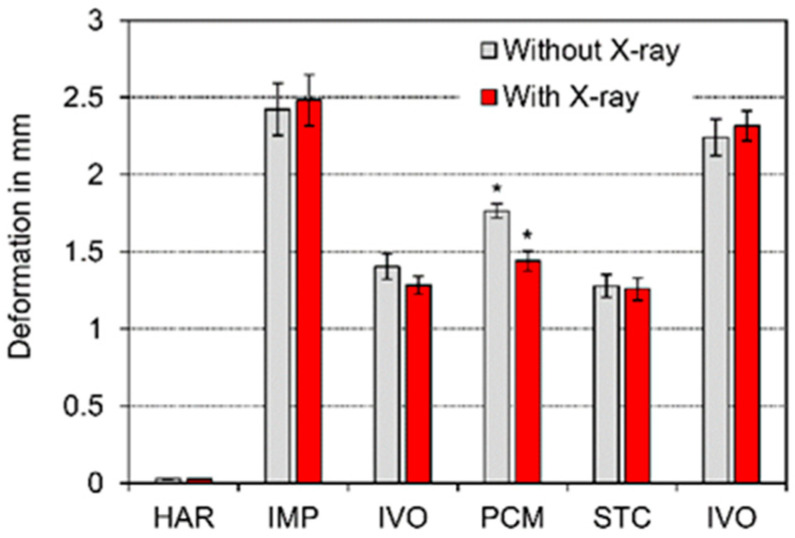
Deformation at the maximum load (*p* < 0.05 → *).

**Figure 5 polymers-13-01271-f005:**
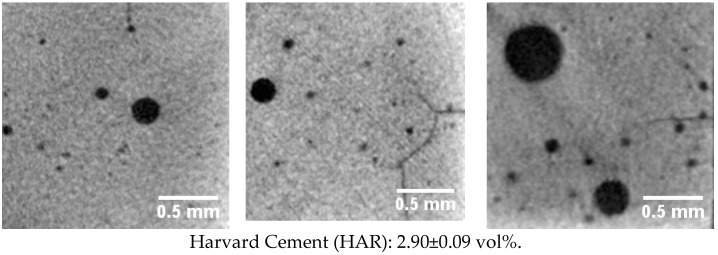
Cross sections (1.87 × 1.87 ≈ 3.50 mm^2^) of three different prisms of each material with their Table 1. 87 × 1.87 × 11.45 mm^3^).

**Figure 6 polymers-13-01271-f006:**
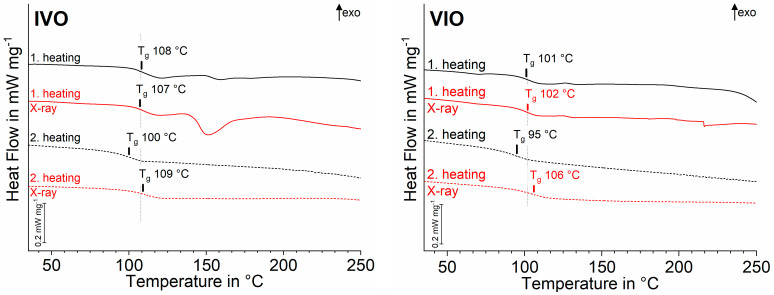
Stacked differential scanning calorimetry (DSC) curves and assigned glass transition temperatures (T_g_, midpoint) for first and second heating of PMMA-based materials exposed to X-ray irradiation during µXCT measurements (red) and unexposed specimens (black).

**Table 1 polymers-13-01271-t001:** Materials investigated in the current study and information on composition and processing type as issued by the manufacturer.

Code	Product Name	Manufacturer	Processing Type	Lot	Composition
HAR	Harvard Cement Normal setting	Harvard Dental International GmbH	Powder Liquid	91706641	Zinc phosphate cement
IMP	IMPRIMO LC Model	Scheu Dental GmbH	3D Printing	4118A	Methacrylate-based resin
IVO	IvoBase Hybrid	Ivoclar Vivadent AG	Powder Liquid	YT1269	Poly(methyl methacrylate) (PMMA)
PCM	PC-ABS Model Material	Stratasys Ltd.	3D Printing	- ^1^	Polycarbonate with acrylonitrile-butadiene-styrene (PC-ABS)
STC	Structur CAD	VOCO GmbH	Milling	1942209	Resin-based Composite
VIO	VITA VIONIC^®^ BASE	VITA Zahnfabrik H. Rauter GmbH & Co. KG	Milling	76380	Poly(methyl methacrylate) (PMMA)

^1^ No medical product.

**Table 2 polymers-13-01271-t002:** Measurement settings employed for the micro X-ray computed tomography (µXCT) analyses.

Measurement Settings		
Specimen	Geometry	12 beams radially arranged on a carbon tube
	Focus–object distance (FOD)	150 mm
X-ray	Voltage	140 kV
	Current	140 µA
Detector	Filter	-
	Exposure time per position	0.999 ms
	Positions	0.45°/360° 800 images
Resolution	Voxel size	8.3 µm edge length 572 µm^3^

**Table 3 polymers-13-01271-t003:** Settings employed for the mechanical testing.

	Units	Mineral-Based HAR	Polymer-Based IVO, PCM, STC, VIO
Support distance	mm	10	20
Approach speed	mm/min	2.00	7.50
Pre-load	N	0.1	1.0
Loading speed	mm/min	0.75	0.75

## Data Availability

The data presented in this study are available in the article.

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
