# Peer review of "Is Micro X-ray Computer Tomography a Suitable Non-Destructive Method for the Characterisation of Dental Materials?"

_polymers, 2021, doi:10.3390/polym13081271_

Round 1
Reviewer 1 Report
Dear Authors,
Your manuscript is well written and its topic has scientific significance. My comments are the following.
Why didn't you use also lower x-ray tube output for example lower voltage values in the μXCT, in order to observe possible changes in the structure of the materials?
Line 235-236: Please remove the phrase in parenthesis.
Author Response
Dear Reviewer 1,
Thank you very much for your input!
“Why didn't you use also lower x-ray tube output for example lower voltage values in the μXCT, in order to observe possible changes in the structure of the materials?
The used voltage for the test setup and samples was normal and not higher than for other comparable samples. We usually irradiated two bending beams (through the rotation, 10 beams were arranged in a circle) and a specimen holder (carbon tube) at the same time. If we reduce the voltage, the grey value distribution in the cross section would be for the same material not constant. It exists mathematic algorithm to reduce this effect, but the noise ratio (see grey value distribution) then increases.
“Line 235-236: Please remove the phrase in parenthesis.”
Yes of course.
Sincerely,
Andreas Koenig

Reviewer 2 Report
Excellent manuscripts. Accept in present form.
Author Response
Dear Reviewer 2,
First of all, we would like to thank you for your review!
Together with the comments from Reviewer 1 and from the editor we try to improve the manuscript in few places.
Sincerely,
Andreas Koenig
